# Genomic Rewilding of Domestic Animals: The Role of Hybridization and Selection in Wolfdog Breeds

**DOI:** 10.3390/genes16010102

**Published:** 2025-01-19

**Authors:** Alžběta Báčová, José Ignacio Lucas Lledó, Kristýna Eliášová, Silvie Neradilová, Astrid Vik Stronen, Romolo Caniglia, Marco Galaverni, Elena Fabbri, Frederica Mattucci, Adam Boyko, Pavel Hulva, Barbora Černá Bolfíková

**Affiliations:** 1Department of Animal Science and Food Processing, Faculty of Tropical AgriSciences, Czech University of Life Sciences Prague, Kamýcká 129, 16500 Prague, Czech Republic; bacovaa@ftz.czu.cz (A.B.);; 2Cavanilles Institute of Biodiversity and Evolutionary Biology, University of Valencia, Catedràtic José Beltrán 2, 46980 Paterna, Spain; joiglu@valencia.edu; 3Department of Zoology, Charles University, Viničná 7, 12843 Prague, Czech Republic; pavel.hulva@natur.cuni.cz; 4Biotechnical Faculty, University of Ljubljana, Jamnikarjeva ulica 101, 1000 Ljubljana, Slovenia; 5Unit for Conservation Genetics (BIO-CGE), Italian Institute for Environmental Protection and Research (ISPRA), Via Cà Fornacetta 9, 40064 Ozzano dell’Emilia, Italyfederica.mattucci@isprambiente.it (F.M.); 6WWF Italy, Via Po 25c, 00198 Rome, Italy; m.galaverni@wwf.it; 7Department of Biomedical Sciences, Cornell University, Ithaca, NY 14853, USA

**Keywords:** wolfdog, hybridization, artificial selection, heterosis, inbreeding

## Abstract

**Background/Objectives**: The domestication of the grey wolf (*Canis lupus*) and subsequent creation of modern dog breeds have significantly shaped the genetic landscape of domestic canines. This study investigates the genomic effects of hybridization and breeding management practices in two hybrid wolfdog breeds: the Czechoslovakian Wolfdog (CSW) and the Saarloos Wolfdog (SAW). **Methods**: We analyzed the genomes of 46 CSWs and 20 SAWs, comparing them to 12 German Shepherds (GSHs) and 20 wolves (WLFs), which served as their ancestral populations approximately 70–90 years ago. **Results**: Our findings highlight that hybridization can increase genetic variability and mitigate the effects of inbreeding, as evidenced by the observed heterozygosity levels in both wolfdog breeds. However, the SAW genome revealed a higher coefficient of inbreeding and longer runs of homozygosity compared to the CSW, reflecting significant inbreeding during its development. Discriminant Analysis of Principal Components and fixation index analyses demonstrate that the CSW exhibits closer genetic proximity to the GSH than the SAW, likely due to differences in the numbers of GSHs used during their creation. Maximum likelihood clustering further confirmed clear genetic differentiation between these hybrid breeds and their respective ancestors, while shared ancestral polymorphism was detectable in all populations. **Conclusions**: These results highlight the role of controlled hybridization with captive-bred wolves and peculiar breeding strategies in shaping the genetic structure of wolfdog breeds. To ensure the long-term genetic health of these breeds, it is recommended to promote adequate and sustainable breeding practices to maintain genetic diversity, minimize inbreeding, and incorporate the careful selection of unrelated individuals from diverse lineages, while avoiding additional, uncontrolled crossings with wild wolves.

## 1. Introduction

The grey wolf (*C. lupus*) represents the first animal species to undergo domestication, an evolutionary process estimated to have occurred approximately 15–40 thousand years ago [1,2,3]. This transformative event led to the emergence of domestic dogs (*C. l. familiaris*). Sometimes even recognized as distinct species [4], wolves and domestic dogs maintain the potential for genetic admixture through interbreeding, resulting in fertile offspring. This phenomenon is attributed to short divergence time, incomplete lineage sorting, and the absence of robust reproductive barriers between the two species. The early domestication was probably primarily driven by selection for tameness [5] that facilitated closer interactions between humans and animals and ultimately might have shaped the evolutionary trajectory of domestic dogs. However, the pathway and the relationship between the first dogs and humans remain unclear [6], and respective hypotheses vary from commensal roles of the ancient dogs to mutual coevolution of humans and canids. Over thousands of years of domestication, and mainly in its late phase, dogs underwent selective breeding for specific functions, morphological characteristics, and other phenotypic traits. The greatest diversification in dog breeds occurred during the Victorian era, driven by intense inbreeding and artificial selection [7,8], which led to the fixation of specific alleles across the genome, increasing genetic homozygosity to stabilize and inherit desired traits [9,10,11]. Efforts to refine and enhance wolf-like traits continued through various breeding experiments across the globe, with notable projects in countries such as China, Russia, the former Czechoslovakia, the Netherlands, and Italy. In these experiments, breeders crossed dogs with wolves with the aim of compensating for drawbacks of inbreeding and overspecialization in modern breeds and creating strong, independent animals with high endurance and sensory abilities, but also retaining trainability and cooperativeness with humans. Several projects, including those in China, Italy, and the former Czechoslovakia, were driven by military objectives. In contrast, in the Netherlands, the experimental breeding of wolfdogs originated as a personal hobby of an individual breeder.

The Saarloos Wolfdog (SAW) originated in the 1930s as part of an experimental breeding project led by Mr. Saarloos. The first successful cross involved a female European wolf of unknown origin donated by the Rotterdam Zoo, and a German Shepherd male from an old Prussian lineage [12]. The breeding process faced significant inbreeding challenges, as Mr. Saarloos repeatedly used the original sire, offspring from the initial pairing, and then probably four additional female wolves, the last of which was introduced in 1963. The exact number of wolves or German Shepherds used is currently unknown, because the stud book and initial pedigree have not been made public. According to Hörter [12], some owners may have introduced other dogs or wolves into the breed between 1940 and 1963 to mitigate inbreeding effects that were visibly reducing body size and robustness of the breed. Inbreeding was further intensified by Mr. Saarloos’s exclusive control over the breeding process, which limited genetic diversity within the line [12]. Today, Saarloos Wolfdogs primarily serve as companion animals, and they remain a relatively understudied breed.

The Czechoslovakian Wolfdog (CSW) is a more extensively studied breed, originating in the 1950s in the former Czechoslovakia with a specific military purpose: to protect national borders during the Cold War. The breed’s foundation pair consisted of a German Shepherd male and a wolf female from the Carpathian population, with additional Carpathian wolf pairings introduced in 1960, 1968, 1974, and 1983, contributing to notable fluctuations in effective population size [13,14]. The CSW genome shows significant allele frequency shifts from both parent populations. These shifts reflect a founder effect and potential hitchhiking, likely due to preferential breeding, or the “popular sire effect”, which amplifies the genetic influence of widely used breeding dogs [13,14,15]. Despite a relatively small population, the breed appears resilient to inbreeding, a phenomenon possibly linked to heterozygote advantage arising from its hybrid origin [13]. This advantage is seen in the form of ’islands of high heterozygosity’ across the genome [14]. Currently, approximately 30% of the CSW genome is derived from wolves, contributing largely to physical traits, such as body posture and facial musculature, as well as to movement, learning abilities, and susceptibility to certain health conditions, including aortic and renal issues [14]. Conversely, genes inherited from dogs influence coat coloration, amino acid metabolism, cognitive function, learning ability, and circadian rhythm [14].

Both wolfdog breeds were officially recognized by the Fédération Cynologique Internationale (FCI)—the SAW in 1981 and the CSW in 1999. Reflecting their hybrid origins, these breeds are genetically distinct from both domestic dogs and wild wolves, though they align more closely with purebred dogs in allele frequency distributions rather than forming an intermediate population [13,14,16]. For both breeds, the appearance is designed to evoke a wolf-like image. Behaviorally, they are loyal to their owners; however, the SAW is notably more reserved, exhibiting avoidance of unfamiliar situations. Both breeds have well-developed jaws and teeth, while specific aesthetic traits differentiate them further. The CSW’s eyes are required to be amber, whereas the SAW’s eye color should be yellow, with brown considered undesirable. The SAW has medium-sized ears that are larger than those of the CSW. Coat color also varies, with CSWs displaying yellowish-gray to silver-gray coats with a distinctive light facial mask, while SAWs range from light to dark shades of black or brown-tipped game-color (wolf-gray) with markings from light creamy white to pure white. Although facial markings are present in both breeds, the SAW is required to display significant, clearly distinctive masks. The markings are genetically influenced by mutations in *MC1R* [15,17].

In this study, we aimed to investigate the genetic diversity and inbreeding levels in Saarloos and Czechoslovakian Wolfdogs and their ancestral populations—German Shepherds and wolves. We also aimed to explore their genetic relationships and to assess the demographic trends of wolfdogs in the context of their unique breeding history.

## 2. Materials and Methods

### 2.1. Genotypes and Data Filtration

For the present study, we combined available SNP data from the published scientific literature [14,18,19] with SNP data generated by Embark Veterinary, Inc. (Boston, MA, USA; EMBARK; Table 1). In all cases, the CanineHD BeadChip (Illumina, Inc., San Diego, CA, USA) or a version of it had been used to genotype dogs or wolves in a number of loci that varied between 137,978 and 216,184. We used plink 1.9 [20] to carry out the following procedure. First, data sets were filtered to keep only samples of interest, as shown in Table 1. Then, we updated the map information (SNP coordinates) of the EMBARK data set, to match the CanFam 2.0 coordinates, common to all other data sets. We removed 50,226 SNPs from this data set that are only typed in EMBARK’s own version of the beadchip. Then, we sequentially merged the data sets, making sure there were no conflicts among them. Only the oldest data set, by Vaysse et al. [19], had SNPs named differently and SNPs where alleles were represented by the nucleotides in the other strand. Based on the positions and similarity of the SNP identifiers, we corrected the 19,505 SNP names in the Vaysse et al. [19] data set. Using plink’s flip option [21], we reversed the strands of 61,307 SNPs in the same data set, to match the pair of nucleotides reported as alleles. SNPs where the two alleles are either A and T or C and G cannot be spotted as reversed in one data set with respect to another. Thus, we removed those from analysis. We also updated the map coordinates of all SNPs to the CanFam 6.0 version of the dog reference genome using the liftOver program (https://genome-euro.ucsc.edu/goldenPath/help/hgTracksHelp.html#Liftover, accessed on 6 November 2024). Only 2307 SNPs were lost in this step. Finally, we removed SNPs where more than 6 samples had missing genotypes. In the end, we retained 114,358 autosomal SNPs from 98 samples. The overall genotyping rate was 0.9898. The total data set consisted of 46 CSWs, 20 SAWs, 12 GSHs, 7 Carpathian wolves [18], and 13 wolves of unknown origin [19].

### 2.2. Population Characteristics and Homozygous Regions in the Genome

Observed heterozygosity (H_O_), expected heterozygosity (H_E_), and polymorphic sites in population were calculated using the col.summary function within the snpStats package in R. The pairwise F_ST_ parameter is widely used for estimation of genetic differentiation among sampled populations [22,23]. The parameter was computed using the Fst function provided by the package.

The principle of runs of homozygosity (ROHs) analysis consists of detecting extensive homozygous regions and serves as an estimation of the current inbreeding level and an indicator of recent or ancient inbreeding events. In general, this analysis assesses the homozygous portion of the genome. The detectRUNS R package was used, with a window size of 50 SNP loci, and a minimum number of 20 homozygous SNPs per run. We estimated the mean percentage coverage per individual for 5 classes based on their length (0–2 Mbp, 2–4 Mbp, 4–8 Mbp, 8–16 Mbp, and >16 Mbp) and the average length of the run per class. Furthermore, the inbreeding coefficient derived from ROHs (F_ROH_) was estimated as the proportion of the genome present in ROHs over the overall length of the genome covered by the analyzed SNPs.

### 2.3. Population Structure and Clustering

To investigate the ancestral composition of the submitted population, we used ADMIXTURE 1.3.0 [24], which employs a maximum likelihood approach to estimate the proportions in which a number K of diverged populations mixed recently. The analysis was conducted using an autosomal data set to assign each individual to its respective population and to identify potential instances of introgression among the populations under investigation. The number of populations (K) tested ranged from 2 to 5. ADMIXTURE runs with unsupervised clustering under default settings. The optimal predictive accuracy was assessed through cross-validation error analysis (CV), with the lowest error value indicating the most supported number of clusters [24]. The data set was subsequently analyzed using Discriminant Analysis of Principal Components (DAPC) through the function dapc() in the adegenet R package [25]. The DAPC analysis is a multivariate method that classifies high-dimensional data into clusters and provides an insight into the structure of the submitted sample [26]. First, a DAPC was run retaining 30 PC axes, then the function optim.a.score was run to identify the optimal number of axes as recommended in Miller et al. [27]. The DAPC was then re-run with the optimal number of 7 retained PC axes.

### 2.4. Demographic Trends

The demographic trends of SAWs and CSWs represented by effective population size (*N_E_*) were reconstructed using the equation E(r^2^) = [1/(1 + 4 *N_E_ c*) + 1/n], where r^2^ is the squared correlation of genotypic association between autosomal SNPs (representing the extent of LD), *c* is the genetic distance between SNPs in Morgans (assuming 100 Mb = 1 Morgan), and 1/n is the correction factor for small sample sizes [28,29]. Using the equation, the demographic changes that occurred 1 to 30 generations ago were estimated. Considering a dog generation interval of 3 years [30], that corresponds to 3–90 years in the past, the whole history of both breeds was covered. We hypothesized that the effective population size (*N_E_*) would increase with the introduction of additional wolves, followed by a gradual decline. This decline occurs because only a subset of individuals contributes to breeding, and the time elapsed since the establishment of the breeding population is insufficient to accumulate new genetic variants. The current population size does not allow for substantial genetic variation to accumulate, as demonstrated by the calculation: *p* = mu × n. gen. × *N_E_* = 1 × 10^−8^ × 20 × 20,000 = 0.004.

## 3. Results

### 3.1. Population Characteristics and Homozygous Regions in the Genome

The expected heterozygosity (H_E_) for SAWs reached the value of 0.237 and for observed heterozygosity (H_O_) of 0.240. In CSWs, the value of H_E_ was estimated to be 0.243, and H_O_ of 0.267. In comparison, the GSH had estimated H_E_ of 0.239 and H_O_ of 0.238. According to the number of polymorphic SNPs within a population, the highest variability was observed in wolves (101,309) and the lowest in purebred GSHs (84,291). Both hybrid breeds, CSWs and SAWs, reflected the same number of SNP variants with minor differences (Table 2).

Genetic differentiation assessment using the fixation index (F_ST_; Table 3) revealed the highest level of divergence between the GSH and WLF (0.211), consistent with general expectations. The CSW reflected the lowest genetic differentiation from the GSH (0.071), while showing a higher degree of differentiation from wolves (0.159). The SAW demonstrated the highest level of differentiation from the WLF (0.182) and the lowest genetic divergence from the CSW (0.127).

The analysis of runs of homozygosity (ROHs) conducted on the autosomal data set detected the highest proportion of two long run classes (8–16 Mb: 17.9%, >16 Mb: 11.6%; Table 4) in the SAW with average lengths of 11.2 Mbp and 26.9 Mbp, implying inbreeding events in recent history of the breed (Appendix A). The trend in wolves seems to be the opposite, having a significantly higher proportion of the shortest runs (0–2 Mbp: 58.5%) and the lowest proportion of the longest homozygous regions (>16 Mbp: 3.1%), and therefore matched typical patterns observed in wild populations. The CSWs possess the second highest proportion of the class with the longest runs of homozygosity (8.5%) and the highest proportion of medium-long runs (22.3%). Figure 1 provides a visual summary of the results. The estimated level of inbreeding deduced from runs of homozygosity (F_ROH_) varied from 0.123 in the WLF to 0.376 in the SAW (Table 4). The GSH (F_ROH_ = 0.357) reached slightly higher values than the CSW (F_ROH_ = 0.354).

### 3.2. Population Structure and Clustering

The hybrid breeds (CSW and SAW) are clearly distinguished from both parental populations (GSH and WLF; Figure 2) and formed clearly bounded groups. Linear Discriminant 1 positions both wolfdog breeds on opposite sides of the axis, while Linear Discriminant 2 suggests their certain proximity. The genetic variability within SAWs is considerably higher than in CSWs. The final number of incorporated axes is 7 (Appendix A). ADMIXTURE analysis revealed the lowest CV (0.4874) at K = 4, clearly recognizing all four studied groups—SAW, CSW, GSH, and WLF—with very limited deviations (Figure 3). Mean estimated membership (Q value, Table 5) to the population was 0.99 (±0.02) for WLFs, 0.96 (±0.04) for SAWs, 0.99 (±0.01) for GSHs, and 0.97 (±0.04) for CSWs.

### 3.3. Demographic Trends

The demographic trajectories estimated from LD well reflects the history of the two breeds, which experienced continuous population declines beginning about 20–25 generations ago, ranging from a maximum of 1840 individuals in 1943 to a minimum of 24 individuals in 2021 for the SAW, and from a maximum of 1076 individuals in 1955 to a minimum of 65 individuals in 2021 for the CSW. In particular, the estimated demographic trajectories showed a first *N_E_* peak in SAWs at the beginning of the 1940s and in the CSW in the mid 1950s (Figure 4), likely corresponding to the first wolf × dog crossings during the origin of the breeds, followed by a progressive decline interspersed with minor peaks possibly corresponding to subsequent wolf or dog additions [16]. Overall, the estimated *N_E_* was higher for the CSW than for the SAW.

## 4. Discussion

In this study, we analyzed the genome-wide profiles of 46 Czechoslovakian Wolfdogs (CSWs) and 20 Saarloos Wolfdogs (SAWs) to investigate the possible effects of hybridization and breeding management on their genomic make-up. These hybrid breeds provide a unique opportunity to examine the genomic interactions associated with interspecies hybridization in detail. As reference populations, we included the genome-wide profiles of 12 German Shepherds (GSHs) and 20 wolves (WLFs), which represent a good available proxy of the ancestral populations involved in the creation of both hybrid breeds approximately 70 to 90 years ago. Our findings indicate higher inbreeding levels in the SAW compared to the other studied canid groups. Furthermore, all four groups exhibit clear genetic differentiation, with signs of an ancestral polymorphism rather than recent gene flow.

The selection for particular traits during human–dog coevolution followed by creation of modern dog breeds approximately 200 years ago has led to the extensive phenotypic variability currently observed in dogs [31]. In combination with the initial domestication phase, the creation of modern breeds is considered a parallel series of major bottleneck events that significantly increased homozygosity in domestic dogs [32]. Reduced variability within dog breeds is primarily driven by the founder effect, which limits the genetic variability at the beginning of the breed creation. Later, breeding practices such as preferential use of some males (popular sire effect), repeated breeding of the same parents, and/or frequent breeding of closely related individuals contributed to the loss of genetic variability. Moreover, the loss of some alleles can also be caused by genetic drift [33,34,35,36,37,38,39,40]. The options to increase the genetic variability of an inbred population are limited. Among the most effective approaches, there is hybridization, namely crossings with individuals from a different taxon. Hybridization, as a natural phenomenon, can increase genetic variability and accelerate speciation by introducing novel genomic combinations that drive evolutionary divergence [41]. In livestock production, hybridization is commonly utilized to exploit heterozygote advantage, or hybrid vigor, which enhances performance traits, leading to improvements in both yields and profitability [42,43]. The use of wild species to augment traits in their domestic counterparts has also been documented, for example, in *Bos* species, with the crossbreeding of bison (*Bison* sp.) and domestic cattle (*Bos taurus*) [44]. Similarly, wild boars (*Sus scrofa*) have been used to improve the fitness of domestic pigs (*S. s. domesticus*) [45].

Our genome-wide analysis of the two wolfdog breeds (CSW and SAW) indicates that the levels of observed (H_O_) vs. expected heterozygosity (H_E_) are sufficient, whereas the values for the GSH show a deficit in heterozygotes (Table 2). This is in line with the fact that the modern dog breeds suffer from increased homozygosity in their populations [38,39]. Their stud books are often closed, since the breed establishment at the beginning of the 19th century had none or little allele migration from other breeds and/or populations. Along with genetic drift, selection pressure, and often insufficient management by the breeders, the loss of alleles is inevitable, resulting in increasing homozygosity. In the case of the wolf sample used in our study, the H_O_ values were estimated lower than in the other studied populations and did not reach the value of H_E_, showing lack of heterozygotes. We expect this discrepancy might be influenced by the unbalanced numbers of samples in studied populations, Wahlund effect, ascertainment bias, and/or a limitation of the array used for the genotyping to capture the true genetic variability of the wolf genome [46].

The ROHs distributed across the genome provide valuable insight into the hybrid breed’s historical development [47,48]. The selection pressure within both wolfdog breeds is comparable due to the similar breeding conditions and requirements and, therefore, our findings suggest that the SAW experienced significant inbreeding in recent history, as evidenced by the elevated levels of medium-long and long ROHs compared to the CSW. While the exact number of wolf ancestors remains uncertain, unofficial records indicate that only one GSH was utilized twice in the SAW population [12]. In contrast, according to Hartl and Jehlička [49], the CSW’s genesis involved four wolves, although the precise number of GSH individuals used remains unclear. Although the level of previously discussed heterozygosity indicates that hybrid breeds benefit from heterosis, it is also important to draw attention to the coefficient of inbreeding (F_ROH_), clearly indicating that the SAW genome shows more regions identical by descent than the CSW or GSH, further confirming that the history of the SAW breed was accompanied by inbreeding events severely impacting the current genome. However, it is important to consider that the SAW is approximately 6 to 10 generations older than the CSW.

Discriminant Analysis of Principal Components (DAPC) suggests a disruptive process between the wolfdog breeds. The CSW demonstrates a closer genetic distance to the GSH than the SAW, supported by DAPC and F_ST_ results, implying a scenario where a higher number of GSHs were used in the development of the CSW than in the SAW. Furthermore, the position of SAWs and CSWs in DAPC (Figure 2) suggests significant differences in the relationship of both hybrid breeds towards the analyzed wolves. The wolves used during the process of CSW creation belonged to the Carpathian population that geographically spans mostly across Czechia, Slovakia, Poland, Romania, and Ukraine [50,51,52,53]. In our study, however, the CSW’s genetic distance to the WLF cluster seems smaller than between the WLF and SAW (Figure 1), suggesting a distinct wolf population has been used for SAW creation.

Despite the hybrid origins of the SAW and CSW, the maximum likelihood method implemented in ADMIXTURE [24] clearly distinguished all three dog breeds and wolves, irrespective of the over-representation of CSW individuals (n = 46). Although both wolfdog breeds share the same ancestral species, the population membership of wolves used in the creation of the SAW remains unclear. In addition, the SAW stud books are not public, and non-registered breeding events cannot be excluded. Nevertheless, there is clear separation between the wolfdog breeds, likely due to the differences in the foundation stock, together with different selection pressures and genetic drift.

Demographic history reconstructed by effective population size (*N_E_*) trajectories to the past provided insight into the history of both breeds. The *N_E_* peaks indicate possible outcrossing events either with other wolves or dogs. Our results for CSWs roughly correspond to the official records of crossings with wolves, also confirmed by Caniglia et al. [14]. The highest peak in SAW points to the first crossing of a wolf and a German Shepherd during the late 1930s. The following peaks of increased levels of *N_E_* estimates suggest further substantial crossings in the 1950s, 1960s, and 1980s. However, peak timings should be interpreted with caution because the results might be biased by over- or underestimation of the generation interval, variations in local recombination rate across the genome, the reported ages of dogs during sample collection, or the dates of sample collection. Although the origin and number of wolves used in the breeding of SAWs remains unclear, the fluctuation of effective population size through time provides more evidence of severe inbreeding that the breed suffered from during its development.

## 5. Conclusions

Our research provides insight into how controlled and deliberated anthropogenic hybridization can positively impact the genome of domesticated species. Rewilding, the introduction of alleles from wild relatives into the gene pool of domestic animals, has previously been applied in domestic pigs [45] and cattle [44]. In this study, we focused on its application in the gene pool of domestic dogs. Although both the Czechoslovakian and Saarloos Wolfdogs originated from equivalent ancestral populations, our findings demonstrate evidence of ongoing disruptive selection pressures. Our results of observed heterozygosity (H_O_) suggest that both wolfdog breeds have benefited from heterozygosity gained through hybridization with captive-bred wolves. However, frequent breeding of related individuals may outweigh the advantages of hybridization, as documented by inbreeding levels (F_ROH_). The genome of the Czechoslovakian Wolfdog exhibits greater heterozygosity and less evidence of inbreeding compared to the Saarloos Wolfdog, likely due to differences in breed management and potentially a broader spectrum of breed founders. In contrast, the Saarloos Wolfdog’s genome-wide composition shows elevated inbreeding levels, with longer runs of homozygosity, indicative of a more restrictive genetic history. These findings emphasize the significant role of hybridization with captive-bred individuals in boosting the genetic variance in the early stages of breed formation while also highlighting the detrimental effects of common breeding practices, such as limited founder stock and frequent use of related individuals, on long-term genomic health.

## Figures and Tables

**Figure 1 genes-16-00102-f001:**
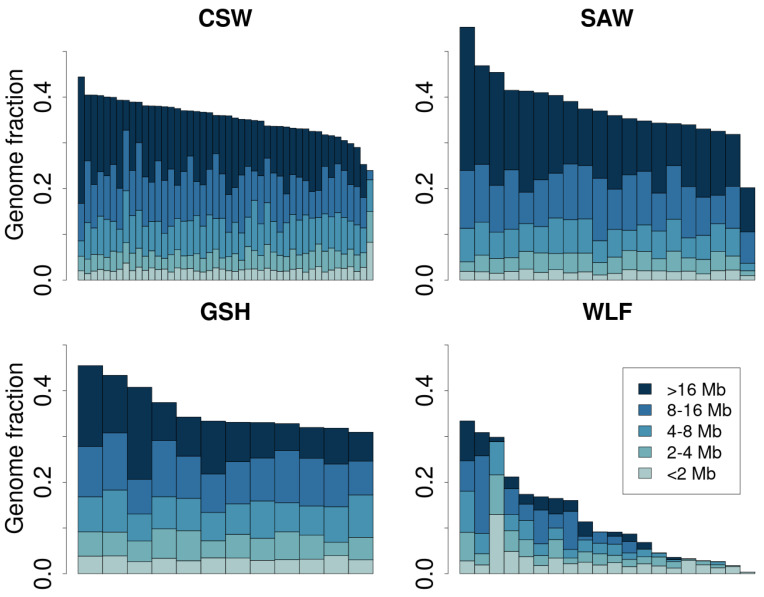
Genome-wide distribution of ROH length classes for each individual of the studied breeds: Czechoslovakian Wolfdogs (CSWs), Saarloos Wolfdogs (SAWs), German Shepherds (GSHs), and wolves (WLFs).

**Figure 2 genes-16-00102-f002:**
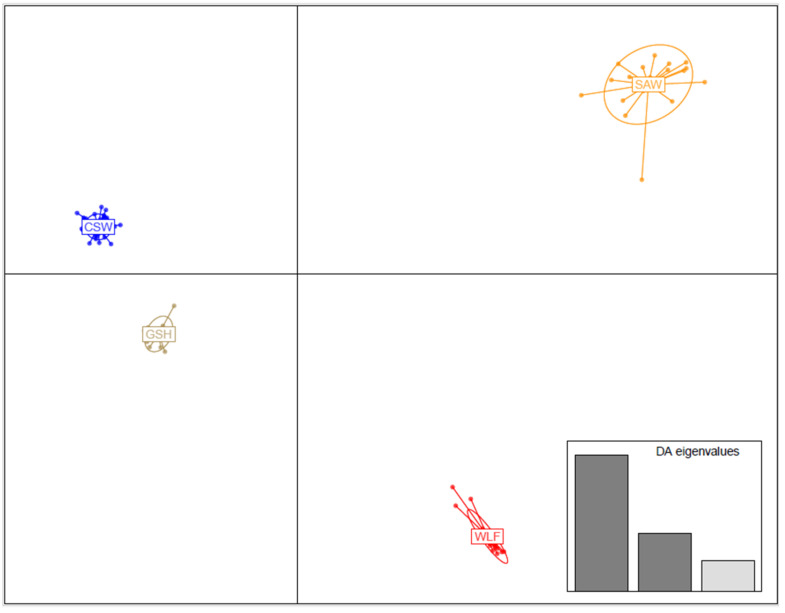
Discriminant Analysis of Principal Components (DAPC) showing the relationships among the 4 studied groups—Saarloos Wolfdogs (SAWs), Czechoslovakian Wolfdogs (CSWs), German Shepherds (GSHs), and wolves (WLFs).

**Figure 3 genes-16-00102-f003:**
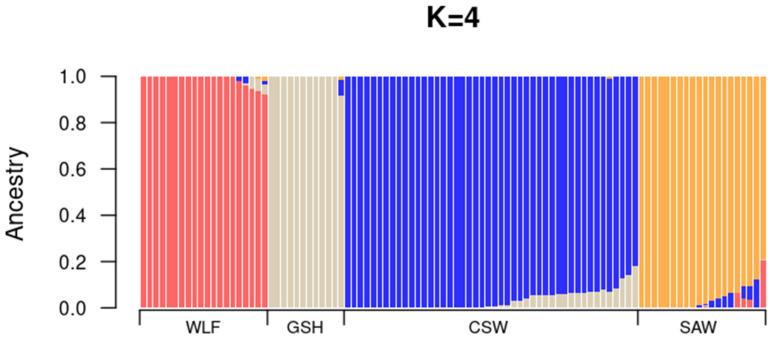
Bar plot showing the estimated population memberships (at K = 4) inferred from ADMIXTURE analysis. Each vertical bar represents an individual, partitioned into colored segments. The length of each segment corresponds to the individual’s coefficient of membership (*q*i) in one of the four clusters: SAWs (Saarloos Wolfdogs), CSWs (Czechoslovakian Wolfdogs), GSHs (German Shepherds), and WLFs (wolves).

**Figure 4 genes-16-00102-f004:**
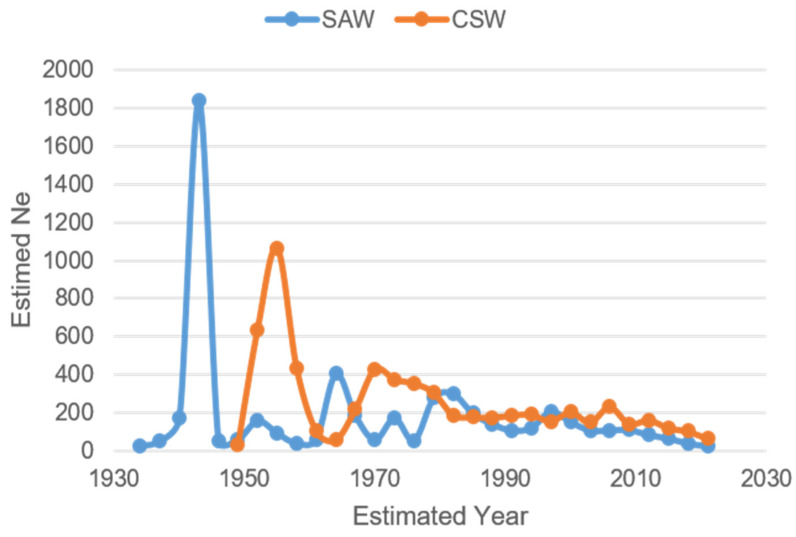
Trajectories of *N_E_* in both wolfdog breeds tracked to the past. Saarloos Wolfdogs (SAWs), Czechoslovakian Wolfdogs (CSWs).

**Table 1 genes-16-00102-t001:** Number of samples of each kind available (and actually used) and number of single nucleotide polymorphisms (SNPs) genotyped in each data set: Saarloos Wolfdog (SAW), Czechoslovakian Wolfdog (CSW), German Shepherd (GSH), wolves (WLFs).

Source	SAW	CSW	GSH	WLF	SNP
EMBARK	18 (18)	32 (32)	0	0	216,184
Caniglia et al. [14]	0	12 (12)	0	0	173,662
Stronen et al. [18]	0	0	0	59 (7)	137,978
Vaysse et al. [19]	2 (2)	3 (2)	12 (12)	15 (13)	174,810
**Total**	**20**	**46**	**12**	**20**	**114,358**

**Table 2 genes-16-00102-t002:** Population characteristics of Saarloos Wolfdogs (SAWs), Czechoslovakian Wolfdogs (CSWs), German Shepherds (GSHs), and wolves (WLFs). Expected heterozygosity (H_E_), observed heterozygosity (H_O_), within-group polymorphic sites, and % of missing genotypes.

	H_E_	H_O_	Within-Group Polymorphic Sites	Missing Genotypes (%)
SAW	0.237	0.240	86,920	0.05
CSW	0.243	0.267	86,908	0.19
GSH	0.239	0.238	84,291	0.07
WLF	0.272	0.227	101,309	4.43

**Table 3 genes-16-00102-t003:** Estimation of genetic distances among studied populations measured as fixation index (F_ST_). Saarloos Wolfdogs (SAWs), Czechoslovakian Wolfdogs (CSWs), German Shepherds (GSHs), and wolves (WLFs).

	CSW	SAW	GSH
**SAW**	0.127		
**GSH**	0.071	0.134	
**WLF**	0.159	0.182	0.211

**Table 4 genes-16-00102-t004:** Percentage representations of individual classes of runs of homozygosity and inbreeding coefficients F_ROH_ estimated from ROHs in the genomes of Saarloos Wolfdogs (SAWs), Czechoslovakian Wolfdogs (CSWs), German Shepherds (GSHs), and wolves (WLFs).

Mbp	SAW	CSW	GSH	WLF
0–2	27.2%	32.7%	36.6%	58.5%
2–4	22.9%	22.3%	25.9%	22.6%
4–8	20.3%	21.4%	18.4%	11.4%
8–16	17.9%	15.0%	13.0%	4.4%
>16	11.6%	8.5%	6.1%	3.1%
F_ROH_	0.376	0.354	0.357	0.123

**Table 5 genes-16-00102-t005:** Average Q values of assessed memberships in ADMIXTURE and their standard deviations (SDs). Saarloos Wolfdogs (SAWs), Czechoslovakian Wolfdogs (CSWs), German Shepherds (GSHs), and wolves (WLFs).

Q Value	MEAN	SD
SAW	95.95%	0.043
CSW	96.75%	0.035
GSH	99.29%	0.013
WLF	98.73%	0.019

## Data Availability

The genotypes of Carpathian wolves used in this study are available in Stronen et al. [18]: https://datadryad.org/stash/dataset/doi:10.5061/dryad.p6598 (accessed on 28 June 2022). The genotypes of the rest of the wolves (13), 12 German Shepherds, 2 Czechoslovakian Wolfdogs, and 2 Saarloos Wolfdogs are available in Vaysse et al. [19]: https://dogs.genouest.org/SWEEP.dir/Supplemental.html (accessed on 28 June 2022). The genotypes of 12 Czechoslovakian Wolfdogs from Caniglia et al. [14] are available on request from the authors and 32 Czechoslovakian Wolfdogs and 18 Saarloos Wolfdogs are available on request at Embark Veterinary, Inc.

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
