# Peer review of "Genomic Rewilding of Domestic Animals: The Role of Hybridization and Selection in Wolfdog Breeds"

_genes, 2025, doi:10.3390/genes16010102_

Round 1
Reviewer 1 Report
Comments and Suggestions for Authors
The manuscript reports on very interesting research. Despite the fact that the authors use genomic data from the gene bank, the genetic characterization and comparison of the studied populations requires serious independent work and modern scientific processing.
At the same time, I would like to improve the quality of the manuscript with some comments.
Line 212: Objective: Both aims are clear, however, following a logical order, I suggest to prioritize the genetic characterization of the two Wolfdogs and the ancestral "species", and then to research the genetic relationship between them. And thirdly, to reveal the demographic trends in the two Wolfdogs.
Line 169-170: I suggest rewording this sentence from two points of view: it suggests that the frequency of ROH is the same as FROH, and that the quotient is not the quotient of two lengths. (proportion of ROH length and genome length)
Line 194: I recommend using “generation interval” instead of “generation time”.
Figure 1: I suggest giving the figure a (new) title (e.g, Distribution of genome fractions ...). The current title is actually a legend. However, it should indicate what is on the horizontal axis.
Table 4 and 5: I suggest merging the two tables. The single row from Table 5 should be included as the last row in Table 4 with a reworded title. Additionally, Table 4 must be written in line 235.
Figure 3: For the sake of general understanding, it is necessary to provide an explanation of the use of the reaching colours.
Table 6: it will be re-numbered for Table 5, also in the text.
Figure 4: Once the pedigrees are available, I suggest also showing the real number of male and female animals by year of birth in the figure.
Line 325 and 365: “Stud Books“ with lowercase initials.
Line 330: “…which is in line with the assumption for wild populations.“ Please explain what exactly you mean by this, with reference.
Line 339-343: The pedigree deficiencies mentioned here contradict lines 281-283.
Line 389-391: This finding is in contrast to the values established for FROH; the values of the two Wolfdogs are practically the same as those of the specifically inbred German Shepherd.
Finally, I suggest comparing the FROH with other breeds/references and incorporating previous genetic evaluations of the Wolfdog (Irish included) into the historical introduction.
Author Response
We sincerely thank the reviewer for his valuable comments and suggestions, which have significantly improved the clarity and quality of our manuscript. The changes are highlighted in the main text.
Here we comment each point of the reviewer:
Line 212: Objective: Both aims are clear, however, following a logical order, I suggest to prioritize the genetic characterization of the two Wolfdogs and the ancestral "species", and then to research the genetic relationship between them. And thirdly, to reveal the demographic trends in the two Wolfdogs.
We have adjusted the aims.
Line 169-170: I suggest rewording this sentence from two points of view: it suggests that the frequency of ROH is the same as FROH, and that the quotient is not the quotient of two lengths. (proportion of ROH length and genome length)
The sentence was reworded as suggested.
Line 194: I recommend using “generation interval” instead of “generation time”.
Adjusted.
Figure 1: I suggest giving the figure a (new) title (e.g, Distribution of genome fractions ...). The current title is actually a legend. However, it should indicate what is on the horizontal axis.
Rephrased
Table 4 and 5: I suggest merging the two tables. The single row from Table 5 should be included as the last row in Table 4 with a reworded title. Additionally, Table 4 must be written in line 235.
Tables were merged and adjusted.
Figure 3: For the sake of general understanding, it is necessary to provide an explanation of the use of the reaching colours.
The title of the figure was adjusted. For clarity, the population membership on axis x was marked in order to ease the readability of the figure.
Table 6: it will be re-numbered for Table 5, also in the text.
Changed
Figure 4: Once the pedigrees are available, I suggest also showing the real number of male and female animals by year of birth in the figure.
Unfortunately, the full pedigrees are not public, but indeed, that would be great analysis, perhaps in the future it will be possible.
Line 325 and 365: “Stud Books“ with lowercase initials.
Changed
Line 330: “…which is in line with the assumption for wild populations.“ Please explain what exactly you mean by this, with reference.
Thank you for this comment, more factors affected the evaluation, which resulted in misleading statements. We avoided the phrase.
Line 339-343: The pedigree deficiencies mentioned here contradict lines 281-283.
We have corrected the result section. Our evaluation was based on overall population size, however, we agree it might be misleading, therefore, we have avoided it completely.
Line 389-391: This finding is in contrast to the values established for FROH; the values of the two Wolfdogs are practically the same as those of the specifically inbred German Shepherd.
The conclusion was clarified, we highlighted the differences in the levels of heterozygosity and inbreeding.
Finally, I suggest comparing the FROH with other breeds/references and incorporating previous genetic evaluations of the Wolfdog (Irish included) into the historical introduction.
Thank you for your suggestion. We would like to clarify that the Irish Wolfhound is a distinct lineage, with no historical or recent genetic admixture involving wolves or wolfdogs. Unlike breeds such as the Czechoslovakian Wolfdog, which is the only relevant comparison and is already addressed in the introduction. We also feel that expanding FROH comparisons to unrelated breeds without historical or genetic connections would likely not contribute additional meaningful insights. Therefore, we prefer to keep the introduction as it is.
Reviewer 2 Report
Comments and Suggestions for Authors
The article submitted to for revision is a interesting analysis of the knowledge and understanding of how genomic is correlated with some hybridization processes and provides insight into how controlled and deliberated anthropogenic hybridization can positively impact the genome of domesticated species. Authors focused on its application in the gene pool of domestic dogs. Both the Czechoslovakian and Saarloos Wolfdogs originated from equivalent ancestral populations, our findings demonstrate evidence of ongoing disruptive selection pressures. Obtained results suggest that both breeds benefit from heterozygosity gained through hybridization with captive-bred wolves. However, frequent breeding of related individuals may outweigh the advantages of hybridization. The genome of the Czechoslovakian Wolfdog exhibits greater heterozygosity and less evidence of inbreeding compared to the Saarloos Wolfdog, likely due to differences in breed management and potentially a broader spectrum of breed founders. In contrast, the Saarloos Wolfdog’s genome-wide composition shows elevated inbreeding levels, with longer runs of homozygosity, indicative of a more restrictive genetic history.
The article written in an appropriate way and all presented data and analyses are presented appropriately. with the highest standards for presentation of the results used. All was presented in short, informative and precise and compact form, facilitating reception and interpretation. The whole study is correctly designed and technically sound good. Used data are robust enough to draw conclusions. All presented methods, tools, software, and reagents are described with sufficient details to allow another researcher to reproduce the results and are potentially applicable for checking. After all, I think, that the paper attract a wide readership and will be interest for a broad audience.
The article is well preserved and organised, and the main topic is original and well-defined. The testing hypothesis and obtained results provide an strong advancement of the current knowledge. Based on this, I think, that the manuscript fit well with the journal scope. Obtained results are interpreted appropriately and are scientifically significant. I’ve found, that all conclusions are well justified and supported by the results. In this matter, hypotheses carefully identified and tested. Used English language appropriate and understandable, I found no stylistic or technical errors. All sources are cited and the literature is well organised. Few technical small issues are marked in the attached, revisited manuscript.
For sure, the work advance strongly the current knowledge. The authors address an important long-standing question with smart experiments. Those research are crucial economic traits in breeding of those dog races. Understanding the molecular mechanisms behind these traits is essential for their genetic improvement. In today's world, where obtaining findings emphasize the significant role of hybridization with captive-bred individuals in boosting the genetic variance in the early stages of breed formation while also highlighting the detrimental effects of common breeding practices, such as limited founder stock and frequent use of related individuals, on long-term genomic health.

Author Response
We sincerely thank the reviewer for his thoughtful and extremely positive feedback on our manuscript. We greatly appreciate the recognition of our work and the valuable insights he has provided. The encouraging comments motivate us to continue contributing to this field, and we are grateful for the time and effort the reviewer has invested in reviewing our study.